# Standard deviation: Standardized bat monitoring techniques work better in some ecosystems

Danny Haelewaters[1,2,3,4]*, Morgan Hughes[3,5,6], José António Lemos Barão-Nóbrega[3,7], Kathy Slater[3,7], Thomas Edward Martin[3,8]

1 Department of Ecology and Evolutionary Biology, University of Colorado Boulder, Boulder, Colorado, United States of America, 2 Faculty of Science, University of South Bohemia, České Budějovice, Czech Republic, 3 Operation Wallacea Ltd, Old Bolingbroke, Spilsby, Lincolnshire, United Kingdom, 4 Biology Centre of the Czech Academy of Sciences, Institute of Entomology, České Budějovice, Czech Republic, 5 Department of Ecology and Environmental Science, Umeå University, Umeå, Sweden, 6 Faculty of Science and Engineering, University of Wolverhampton, Wolverhampton, United Kingdom, 7 rePLANET Ltd, Old Bolingbroke, Spilsby, Lincolnshire, United Kingdom, 8 School of Natural Sciences, College of Environmental Sciences and Engineering, Bangor University, Bangor, United Kingdom

* danny.haelewaters@gmail.com

**Data Availability Statement:** All relevant data are within the manuscript and its Supporting Information files (in particular S1 Table).

## Abstract

Standardized monitoring strategies are often used to study spatial and temporal ecological patterns and trends. Such approaches are applied for many study taxa, including bats (Mammalia, Chiroptera). However, local characteristics of individual field sites, including species assemblages, terrain, climatic factors, and presence or lack of landscape features, may affect the efficacy of these standardized surveys. In this paper, we completed mist-netting surveys for bats in two widely separated field sites, Calakmul Biosphere Reserve (CBR), a Mexican lowland tropical forest, and Krka National Park (KNP), a Mediterranean dry scrub forest in Croatia. Standardized surveys were conducted along predefined transects for six hours. We also completed targeted surveys in KNP that focused on the key bat activity period (the first two to three hours after sunset), with nets being deployed at sites of known or assumed value to bats (independent of predefined transects). We analyzed how survey success differed in standardized surveys between CBR and KNP and between standardized and targeted surveys in KNP. Survey success was measured through three parameters: capture rate = the number of individual bats captured per net hour, inventory rate = the number of unique bat species recorded per net hour, and inventory efficacy = the percentage of known species assemblage recorded per net hour across all surveys. Results for all three parameters indicate that standardized surveys in CBR were vastly more effective than those in KNP (e.g., mist-netting in CBR detected 69.8% of the species assemblage, compared to just 8.3% in KNP), and it was only by employing targeted mist-netting in KNP that meaningful capture rates could be achieved. This study contributes further evidence to discussions around how and when standardized survey methods should be employed, and the alternative approaches that can be taken in ecosystems where generally effective methods underperform.

**Funding:** DH acknowledges support from the Systematics Research Fund of the Linnean Society of London and the Systematics Association. The funders had no role in study design, data collection and analysis, decision to publish, or preparation of the manuscript.

**Competing interests:** The authors have declared that no competing interests exist.

## Introduction

Long-term monitoring is vital for studying ecological patterns and trends on a global scale [1–3]. Standardized monitoring strategies are often recommended to generate the data necessary to evaluate changes in richness, abundance, and distribution of species, to increase the explanatory power of ecological drivers and threats, and to develop conservation strategies [4, 5]. Standardized sampling techniques are used in many different fields, including canopy biology [6, 7], entomology [8], mycology [9], ornithology [10], mammalogy [11, 12], and parasitology [13]. However, an underlying assumption of all these standardized survey methods is that they provide comparable results in all ecosystems.

Assumptions regarding uniform performance of standardized survey methods may be problematic, given that variability in ecological and geographical factors offer challenges and present obstacles to standardized survey efficacy [14, 15]. Different species assemblages may possess different traits that influence their susceptibility to detection [16]. Similarly, terrain may affect the type of equipment that can be used during surveys, climatic factors and seasonality may influence survey effectiveness, and the presence or lack of features attractive to the survey taxa can heavily influence the likelihood of species encounter. For example, Martin and colleagues [10] showed for birds that local characteristics of individual sites significantly influence the efficacy of standardized surveys. To overcome these limitations, bespoke approaches may be more effective. Here we explore the effectiveness of standardized surveys for describing biological communities across multiple sites in another group of organisms: bats (Mammalia, Chiroptera).

With regard to bats, whilst surveillance using passive methods (e.g., acoustic monitoring, infra-red/thermal technology, roost emergence counts) is effective in many situations for meeting the needs of some studies [17, 18], the capture of bats is required to accurately identify cryptic species, to determine demographic and breeding parameters of a sampled population, to obtain samples (DNA, chemical samples, fecal samples, ectoparasites), and to facilitate radio telemetry studies [19–22]. Standard and bespoke equipment exists to facilitate the capture of bats, with mist nets in particular (as opposed to harp traps) being widely used in remote locations owing to their versatility, portability, and affordability [21].

Standardized mist-netting techniques for bats typically utilize mist nets deployed with set spacing for set periods. The placement of nets often focuses on landscape characteristics of interest to bats, including water features (riparian and lentic), roosting sites, and linear features, such as woodland edges [23, 24]. Survey designs aiming to compare sites will standardize the amount and placement of equipment as well as the timing and duration of deployment. Best practices and guidelines for standardized mist-netting for bats are presented by Kunz and Kurta [23], Barlow [24], Walsh and Catto [19], Battersby [21], and Collins [20]. Survey effort for mist-netting has been explored in further detail by Weller and Lee [25] and Hughes and colleagues [26].

In this study, we explored the effectiveness of standardized mist-netting in ecologically and structurally dissimilar ecosystems. We compared the number of individual bats and unique species captured as proxies for trapping efficacy, as well as the proportion of the known bat species assemblage captured based on available data from each site. We also compared the effectiveness of standardized mist-netting with more bespoke, targeted approaches to mist-netting. This will allow for the first quantitative assessment of the performance of standardized mist-netting as a monitoring tool between widely separated and ecologically dissimilar study sites. We formulated two hypotheses: (1) standardized mist-netting is more effective for capturing bats in a landscape with greater heterogeneity and abundant features suitable for commuting (e.g., rivers, forest edges) than in a homogenous landscape; and (2) in homogenous

landscapes, targeted surveys result in greater efficacy in capturing bats compared to standardized surveys.

## Materials and methods

### Ethics and permits

The Ethical Committee for Animal Experimentation at the Faculty of Science of Ghent University approved all capture and sampling procedures under reference EC2021-062 (license number LA1400452). Fieldwork in Croatia was supported by the Croatian Ministry of Environment and Energy (research permit KLASA: UP/I-612-07/20-48/138, URBROJ: 517-05-1-1-20-4, 16.09.2020). Fieldwork in Mexico was supported by the Secretaría de Medio Ambiente y Recursos Naturales and Comisión Nacional de Áreas Naturales Protegidas (research permit SGPA/DGVS/03535/20).

### Description of study sites

Study sites comprised lowland deciduous tropical forest in Mexico and Mediterranean dry scrub-dominated vegetation in Croatia (Fig 1).

Calakmul Biosphere Reserve (CBR) in Mexico (18.60583 N 89.94444 W) is a large (723,000 ha) expanse of lowland deciduous tropical forest in the southern portion of the Yucatán Peninsula (Campeche). CBR is part of the Selva Maya that spans over 10.6 million ha in Mexico, Guatemala, and Belize, making it the largest continuous tract of tropical forest in Mesoamerica [27]. The southern Yucatán Peninsula is characterized by a warm, sub-humid climate with a mean annual temperature of 24.6°C. A precipitation ecocline goes from the northwest (ca. 900 mm) to the southeast (ca. 1400 mm) of the reserve [27], over the 120 km from the north of the reserve to the Guatemalan border [28], significantly influencing forest structure and tree species composition [29, 30]. To date, the species list for bats within the Calakmul study site comprises 53 extant species of which 13 are Phyllostominae, 13 are Stenodermatinae, eight are Vespertilionidae, five are Molossidae, with three species each of Carollinae and Mormoopidae, and two species each of Desmodontinae, Emballonuridae, Glossophaginae, and Natalidae [31]. This list includes recently confirmed records of *Gardnerycteris crenulatum*, *Glossophaga commissarisi*, *Micronycteris minuta*, and *Trinycteris nicefori* (Ivan Samayoa Gomez, Fabián Mora, Gabriel Oviedo, Kathy Slater, unpubl. data) (S1 Table).

Krka National Park (KNP) in Croatia (43.943109 N 15.991765 E) is in an ecotone between the evergreen Mediterranean and the sub-Mediterranean deciduous vegetation zones. The study area predominantly comprises rough Mediterranean scrub characterized by *Carpinus betulus*, *Juniperus oxycedrus*, and *Quercus cerris*. The region experiences a hot-summer climate with temperatures between 5°C in January and 23°C in July. Precipitation averages 1,078 mm per year and falls largely in autumn and winter months [32]. The bat species list for KNP comprises 24 extant species of which 18 are Vespertilionidae, four are Rhinolophidae, and one species each of Miniopteridae and Molossidae [33, 34]. This list includes recently confirmed records of *Myotis aurascens* and *Myotis daubentonii* [35] (S1 Table).

### Standardized surveys

Standardized mist-netting involved the deployment of a set length (site-specific) of 36-mm mesh mist nets during each survey along predefined transects to represent different habitat types based on botanical assemblage. GPS locations of each net were recorded. At both sites, mist-netting was conducted for six hours, starting before or shortly after sunset. Nets were checked every 15 to 30 minutes. Surveys were not carried out during heavy or prolonged rain,

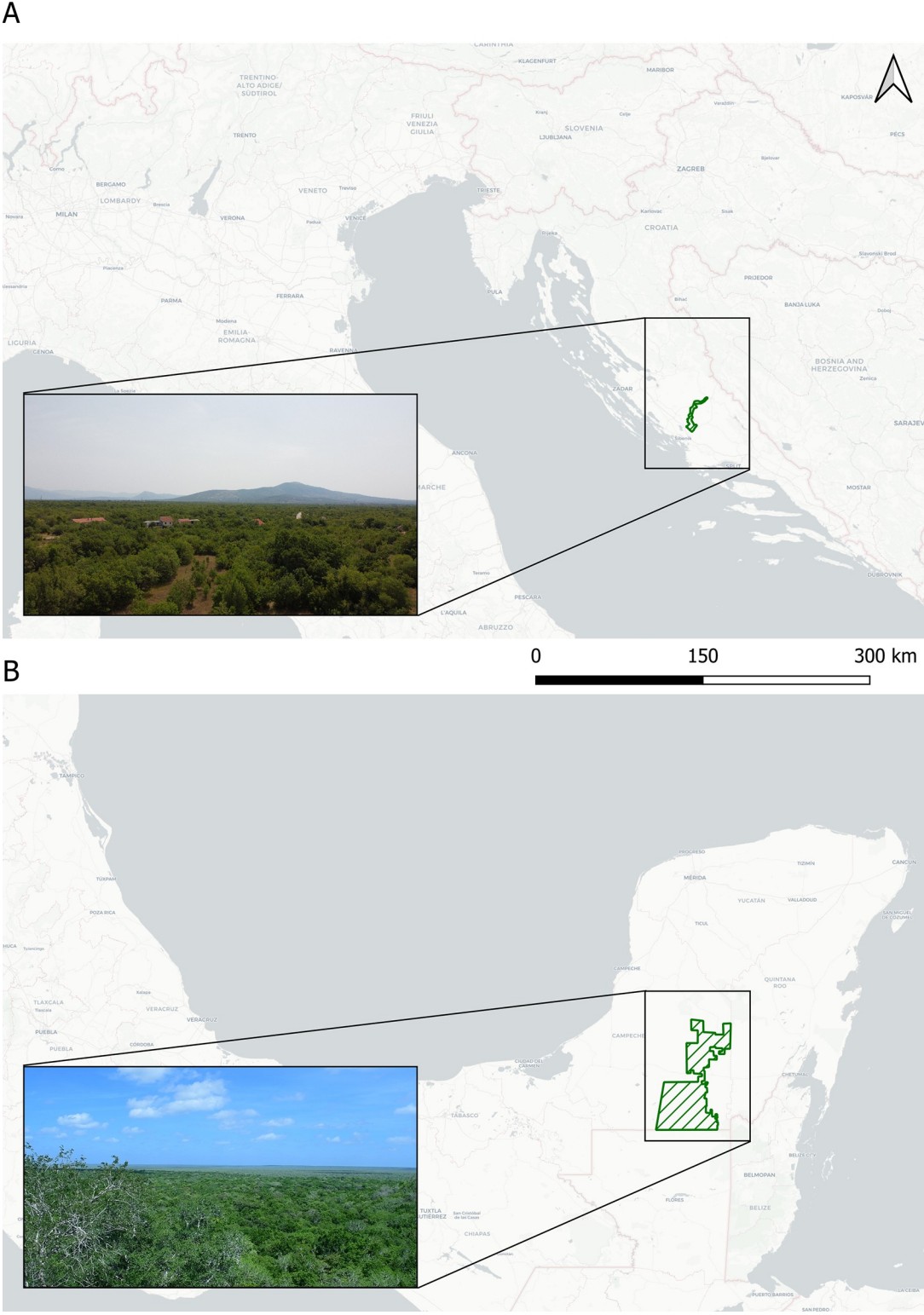

**Fig 1. Map of study sites with inset photos showing main vegetation characteristics.** A. Krka National Park in Croatia. B. Calakmul Biosphere Reserve in Mexico.

nor with heavy wind [36, 37]. No acoustic lures were used for any surveys. Surveys were carried out in the summers (June to August) of 2014 and 2021, at CBR and KNP, respectively.

### Targeted surveys

Targeted surveys (KNP only) broadly followed the same approach as standardized surveys. However, nets were deployed at sites of known or assumed value to bats (e.g., confirmed roosts, mines, caves, water bodies, and riparian corridors) [24], independent of the predefined transects used for the standardized surveys. To facilitate capture of bats at these targeted sites, bespoke methods were used, including the use of a suspended net over a water body [38] and the utilization of a raised net at a mine entrance. Targeted surveys took place for only two to three hours after sunset, as their focus was on emerging bats and post-emergence commuting and drinking/foraging activities. Targeted surveys also did not utilize acoustic lures.

### Processing of bats

Bats were extracted from nets and held in capture bags until processed as per standard protocols [24, 39]. Processing of bats included the identification of species; assessment of sex, age class, and breeding condition of each individual; and the recording of morphometric measurements (body mass and forearm length). In the case of cryptic species, additional measurements (e.g., tragus width, length of tibia) were taken as appropriate. Each bat was marked by fur-clipping or wing punch to ensure that re-captured bats were immediately recognized and released without being subject to repeat processing.

### Quantification of survey effort

As the deployment of nets varied in length among sites, and the length of surveys also varied according to survey type, survey effort was quantified in net hours (NH), defined as 12 m of net deployed for 1 hour [25, 40]. This facilitated the comparison of survey efficacy among sites and between methods.

### Statisical analyses

We examined two measures of survey success: capture rate (hereafter "CR"), being the number of individual bats captured per NH, and inventory rate (hereafter "IR"), being the number of unique bat species recorded per NH. To remove any inherent bias caused by the fact that CBR has a larger bat species assemblage than KNP (53 and 24 species, respectively), we used a third measure of success as inventory efficacy (hereafter "IE"), being the percentage of known species assemblage recorded per NH across all surveys.

We ran a pairwise comparison using Mann-Whitney U tests to explore significant differences in capture rates between the countries. For the KNP data, we also ran Mann-Whitney U tests to determine if the CR, IR, and IE values for standardized surveys were statistically different to those of targeted surveys. All analyses were done in the R language and environment for statistical computing [41]. Figures were produced using ggplot() implemented in the R package *ggplot2* [42].

## Results

The total number of bat individuals captured during standardized surveys in both countries was 1,837, comprising 39 species. CBR represented the greater abundance and species richness, with 1,786 individual bats captured representing 37 species. KNP yielded a meager four individual bats representing two species. When adjusted to account for survey effort in NH,

**Table 1. Survey success of standardized and targeted surveys in Calakmul Biosphere Reserve (CBR), Mexico and Krka National Park (KNP), Croatia.** Columns indicate: number of net hours (NH, 1 NH = 12 m of mist net deployed for 1 hour), number of bats captured (Bats), number of species captured (Species), total known species assemblage for each site (Assemblage) (S1 Table), species captured as percentage of known species assemblage for each site (%), capture rate (CR = number of individual bats captured per NH), inventory rate (IR = number of unique bat species recorded per NH), and inventory efficacy (IE = percentage of known species assemblage recorded per NH across all surveys).

| Site | Type | NH | Bats | Species | Assemblage | % | CR | IR | IE |
|---|---|---|---|---|---|---|---|---|---|
| CBR | Standardized | 926.37 | 1,786 | 37 | 53 | 69.8% | 1.93 | 0.04 | 0.08% |
| KNP | Standardized | 202.50 | 4 | 2 | 24 | 8.3% | 0.02 | 0.01 | 0.04% |
| KNP | Targeted | 30.88 | 47 | 9 | 24 | 37.5% | 1.52 | 0.29 | 1.21% |

the CR values for standardized surveys in CBR and KNP were 1.93 and 0.02, respectively. IR values for CBR and KNP standardized surveys were 0.04 and 0.01, respectively. IE values for standardized surveys at CBR and KNP were 0.08% and 0.04%, respectively. The targeted survey data from KNP comprised 47 individual bats of nine species, representing a CR of 1.52, an IR of 0.29, and an IE of 1.21%. These results are also comparatively shown presented in Table 1. Mann-Whitney U tests (Fig 2) showed significant differences between standardized surveys in KNP and CBR in CR ($W = 6$, $p < 0.001$), IR ($W = 6$, $p < 0.001$), and IE ($W = 6$, $p < 0.001$) values. Within our comparative standardized versus targeted netting datasets for KNP, our Mann-Whitney U tests showed that there was a significant difference between CR values ($W = 6$, $p < 0.002$), IR values ($W = 6$, $p = 0.002$), and IE values ($W = 6$, $p = 0.002$), respectively (Fig 3).

## Discussion

Results demonstrate that standardized surveys in CBR were much more successful compared to standardized surveys in KNP. This was the case for all parameters examined: capture rate, inventory rate, and inventory efficacy. In contrast, the targeted surveys in KNP yielded the highest species inventory rate and species inventory efficacy overall—even higher than those of standardized surveys in CBR. The key to the interpretation of these data lies in the landscape of the study sites included here. A recent paper that compared ornithological survey methods between CNP and lowland forests of Buton Forest Reserves (Buton Island, Indonesia) also found that their efficacy is affected by habitat structure, in addition to composition of local bird communities [10].

We identified three limitations that are associated with our study. First, the number of study sites is limited (two sites: CBR and KNP). This is in part due to the fact that most standardized bat surveys use methodologies that cannot be compared one-to-one. Second, the method selected for our standardized and targeted surveys (i.e., mist-netting) is more likely to capture species and feeding guilds that are more likely to fly within 5 m above ground-level. This excludes high-flying foraging bat species (e.g., in the genera *Eptesicus*, *Nyctalus*, *Tadarida*). While this can be compensated for using acoustic lures [43], these were not employed in any of our surveys. Finally, our site inventories of known species assemblage may not be 100% complete. If an area is relatively understudied, fewer of the species occurring in it will have been recorded, particularly acoustically cryptic species [44, 45]. As a result, with understudied sites, the calculations of survey inventory rate may be overestimated. This factor is, however, unlikely to be a significant factor in our results as multiple years of mist-netting were undertaken at the studied sites, and both sites are ecologically fairly well-studied. In addition, the comparison of standardized versus targeted surveys in KNP is not affected by this factor.

Regardless of these limiting factors, the results of our study are very clear; standardized mist-netting in CBR vastly outperforms the same standardized methods in KNP across all

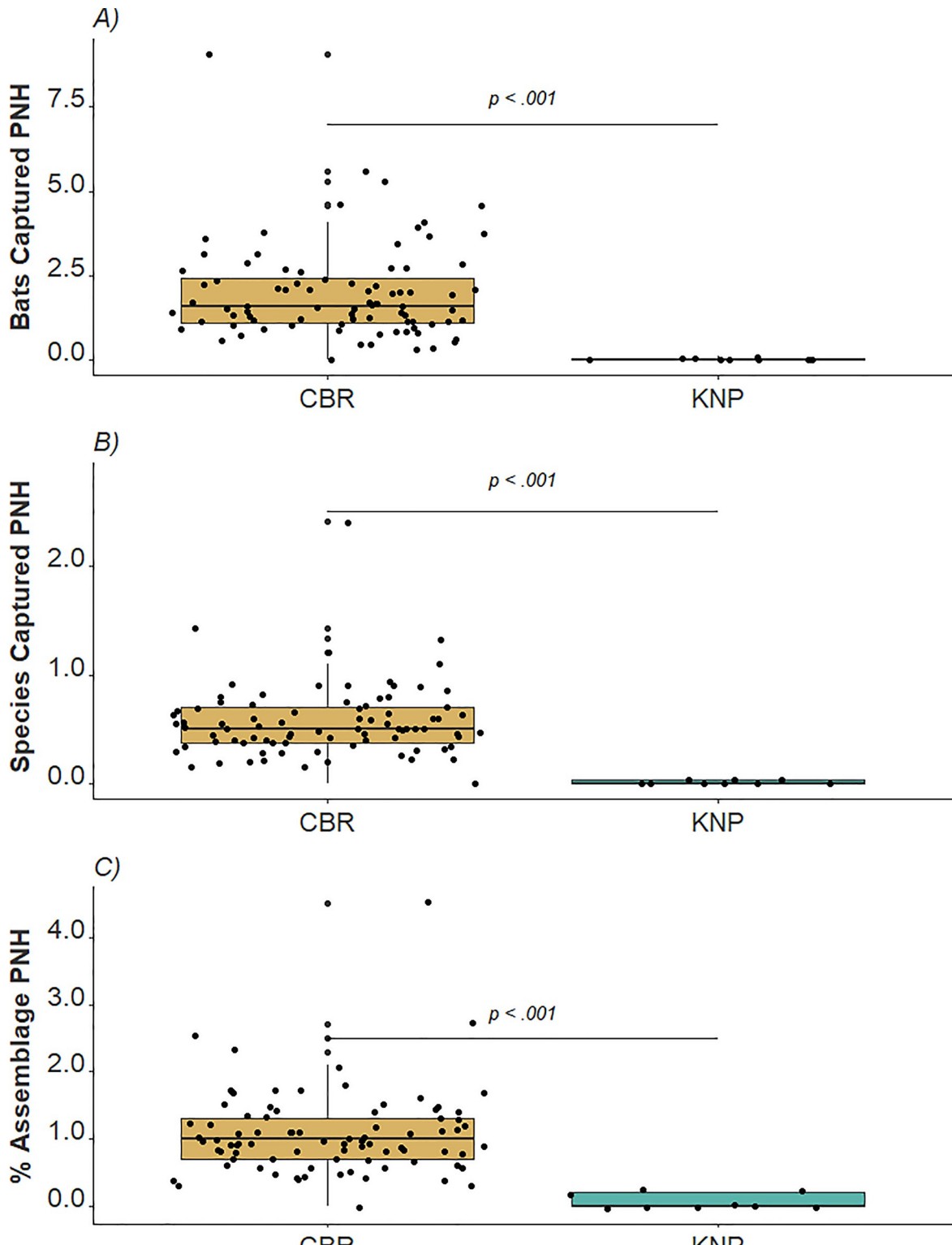

**Fig 2.** Box and whisker plots showing comparative results of Mann-Whitney U tests for A) capture rate, B) inventory rate, and C) inventory efficacy of standardized surveys. Abbreviations: CBR, Calakmul Biosphere Reserve, Mexico; KNP, Krka National Park, Croatia; PNH, per net hour.

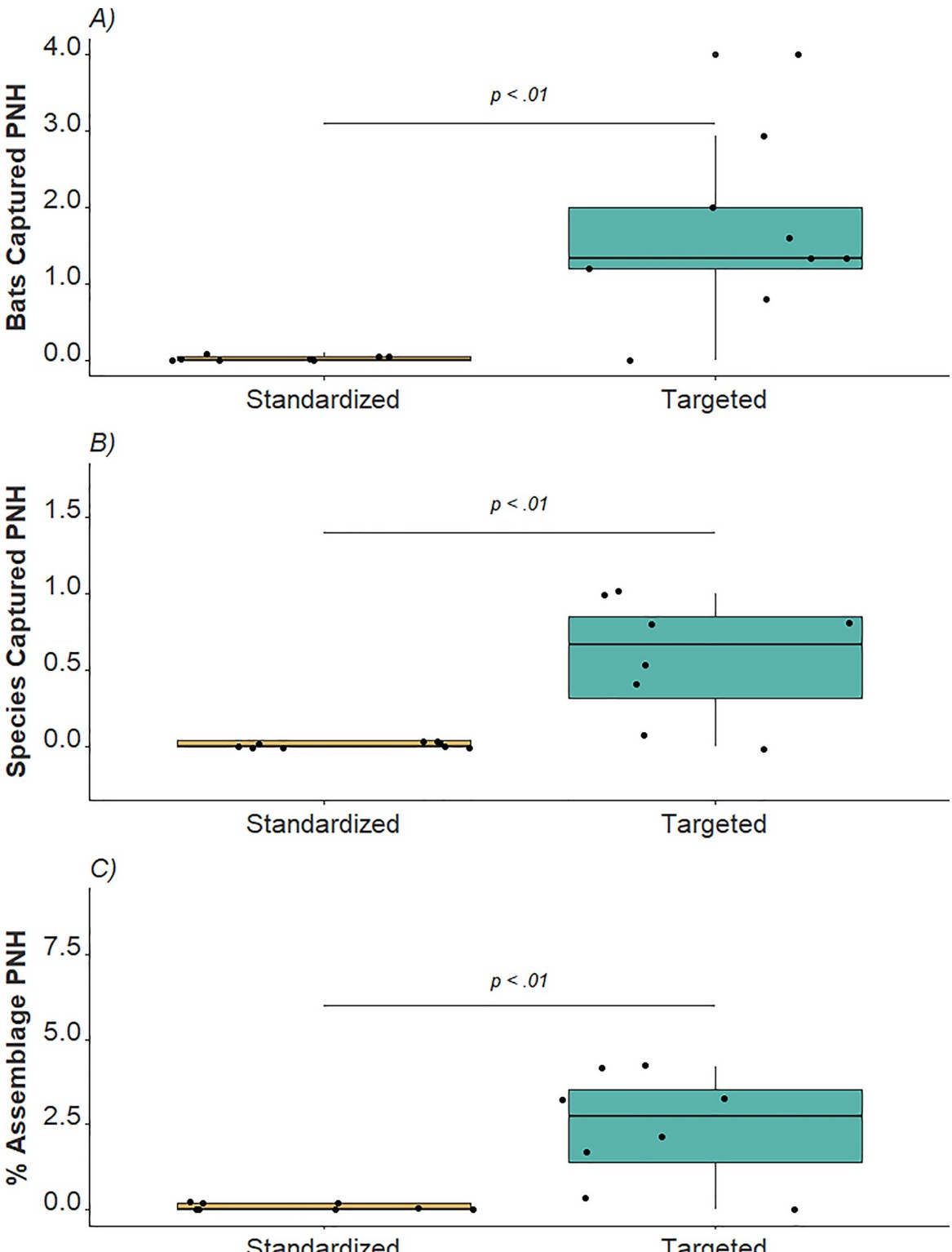

**Fig 3.** Bar plots showing comparative results of Mann-Whitney U tests showed in Krka National Park, Croatia for A) capture rate, B) inventory rate, and C) inventory efficacy of standardized versus targeted surveys. Abbreviation: PNH, per net hour.

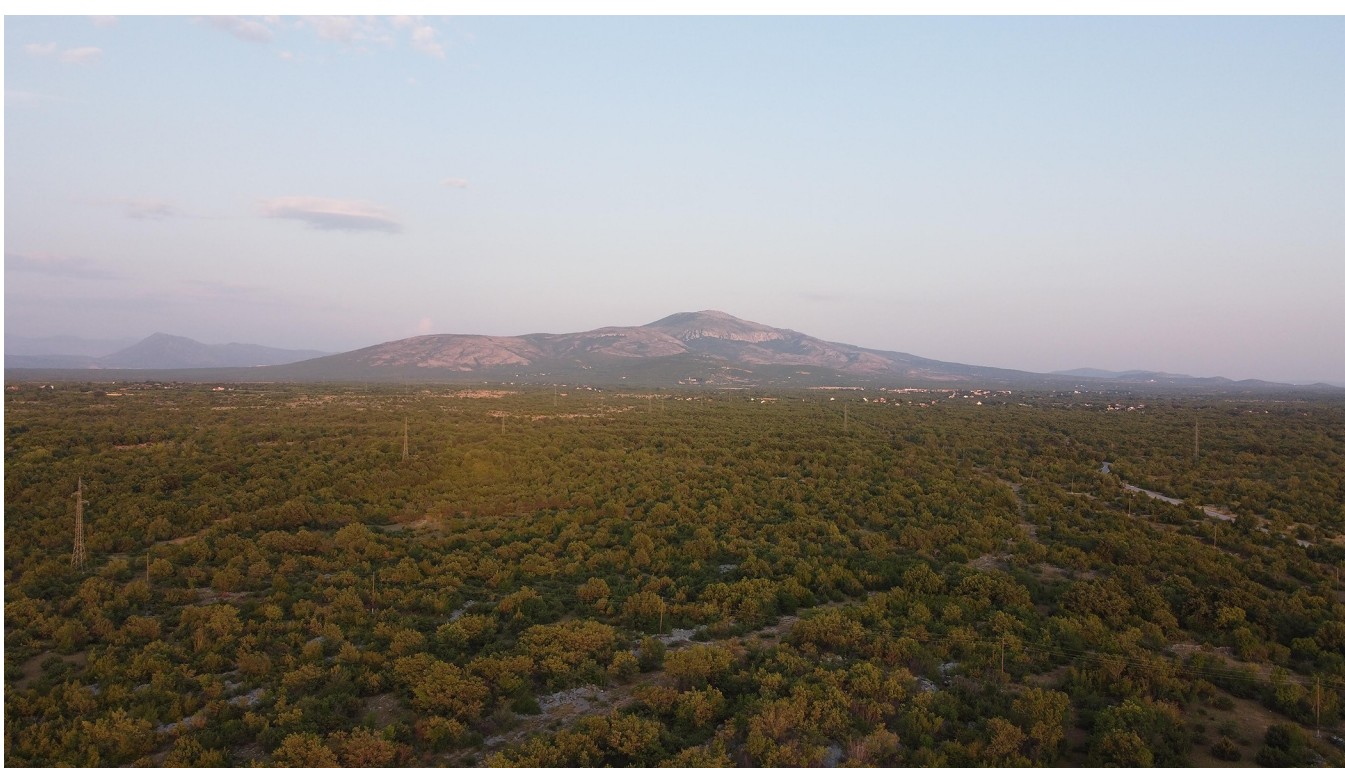

**Fig 4. Homogeneity of Mediterranean scrub landscape at Krka National Park, Croatia.**

measured parameters. In likelihood, these highly disparate results are due to the nature of the landscape in KNP, which differs from CBR in terms of its homogeneity, openness, aridity, and lack of linear features. This is illustrated by the photo taken by a DJI Mavic Mini drone with built-in FC7203 camera (Fig 4). These site-specific characteristics may have contributed to the poor capture and inventory rates in the KNP standardized surveys as the landscape presents few opportunities to place nets along transects in places where bats are likely to be present. The pressures of arid landscapes can affect bat distributions [46] and may force increased bat abundance and species richness at the few water bodies present [47], making even small ponds particularly lucrative, as shown by our data. Similar effects have been observed for avifauna in arid areas, with species richness being negatively associated with distance to water [48].

Whilst the inventory efficacy of targeted surveys at KNP are persuasive, there are behavioral idiosyncrasies at play affecting our data. In the arid karst landscape, all bats must find water daily. Where lentic and riparian habitats are limited (as in KNP), this naturally forces a greater species diversity at fewer existing water sources than might be found at abundant water sources in other landscapes, such as that of CBR. This would likely create a steeper species accumulation curve than in other habitats, and as such the inventory efficacy of targeted surveys at KNP is likely to be skewed to that effect. This idea is somewhat reinforced by the fact that targeted surveys in KNP, whilst much more effective than standardized surveys at KNP, resulted in CR, IR, and IE values that were still lower than those of CBR.

Our results may also be influenced by the overall abundance (or lack) of bats in each of these regions or by the proximity to standardized trapping sites of features of value to bats. In KNP, our standardized trapping sites are located on a plateau above and around the Krka River. All standardized trapping locations in KNP were located within 2 km distance from this

important feature, and between 2.5 and 3.7 km from the Miljacka II cave, where 6,000 bats of eight species roost [49]. This implies that a lack of bat abundance is not a potential cause of low capture rates; rather, it is their distribution across the landscape. Flying is energetically expensive [50], with energetic pressures on bats increasing in summer during lactation and spermatogenesis [51]. Bats tend to fly directly and at great speeds from roosting sites to foraging grounds on their nightly commute to reach optimal energy expenditure [52]. For their commute, they select linear features along which they navigate, such as tree lines, rivers, and forest edges. Bat abundance has indeed been shown to be greater on such linear fly paths [53–55]. The lack of linear features in the KNP landscape greatly reduces the chances that netting will capture bats (particularly as transects are often selected for the monitoring of multiple organismal groups and do not consider the likelihood to encounter bats [56]).

When a new field site is being surveyed for bats for the first time, it is advised to initially use targeted surveys to gain a good first idea of the species assemblage that is present. For example, the first bat inventory in Chucantí Nature Reserve in the remote Darién province, Panama [57] was the result of a targeted survey with mist nets that were positioned over existing trails that were assessed for their potential use by bats as fly paths [55]. Despite a low survey effort (total NH was 34), this survey resulted in 227 bat individuals representing 17 species; the CR value was 6.68 and the IR value was 0.50. While the bat species assemblages of both Chucantí Nature Reserve and the Darién province are unknown, a total of 118 bats are known in Panama [58], resulting in an IE of 0.42%. We note that the local bat species assemblage will be lower given that Chucantí is a cloud forest system, thus with a specific assemblage of bats that occur between 600 and 1480 m altitude [59, 60]. With this in mind, despite the IE value at Chucantí being lower than that for targeted surveys at KNP, it is still higher than those for standardized surveys at CBR and KNP. Also the CR and IR values for the targeted survey at Chucantí are considerably higher than those for standardized surveys at CBR and KNP.

In summary, for the reasons discussed above, our results closely matched the expectations laid out in our hypotheses; (1) that standardized mist-netting performed better in the (more heterogenous) habitats of CBR than in the (more homogenous) landscapes of KNP, and (2) that targeted surveys yielded better results than standardized surveys in the homogenous landscapes of KNP. While it is likely that employing targeted surveys at CBR will significantly increase the capture rate, inventory rate, and inventory efficacy of surveys as it did for KNP in this study and Chucantí Nature Reserve [57], this would need to be weighed against the benefits standardized, comparable datasets being lost [4, 61, 62]. The suitability of standardized survey methods for deployment in different ecosystem types can be hard to judge. It is down to the professional judgment of the individual researcher whether the landscape will lend itself to contributing towards such datasets, or if the results of standardized surveys are likely to be inhibited by landscape, homogeneity, climate, and scarcity of features. This study demonstrates that there is not a "one-size-fits-all" approach to monitoring biodiversity, highlighting the challenges in the development of surveying methods on a global scale. We emphasize that researchers have a responsibility to weigh the costs and benefits of selecting one monitoring approach over another to best effect positive change for species conservation.

## Supporting information

**S1 Table. Overview of all reported species of bats in our study sites.** All reported bat species–including unpublished ones–from Calakmul Biosphere Reserve, Mexico and Krka National Park, Croatia.
(XLSX)

## Acknowledgments

This research was conducted as part of Operation Wallacea's long-term biodiversity monitoring project in collaboration with Biota Ltd in Croatia and Pronatura Peninsula de Yucatan in Mexico. Operation Wallacea provided the logistical support necessary to complete this study. We thank the following individuals for logistical support and assistance with fieldwork: Mark P. Aquilina, Anneka Goeter, Bernice Hyett, Dušan Jelić, Fabián Mora, Gabriel Oviedo, Tommy Saunders, Carmen Sorina, Anna Suvorova, and Oliver Thomas. Sasha Karabasova and Jalal Khan are thanked for permission to use their photographs of Krka National Park and Calakmul Biosphere Reserve, respectively.

## Author Contributions

**Conceptualization:** Danny Haelewaters, Thomas Edward Martin.

**Formal analysis:** Morgan Hughes.

**Investigation:** Danny Haelewaters, Morgan Hughes, José António Lemos Barão-Nóbrega, Kathy Slater.

**Visualization:** Morgan Hughes.

**Writing – original draft:** Danny Haelewaters, Morgan Hughes.

**Writing – review & editing:** Danny Haelewaters, Morgan Hughes, Thomas Edward Martin.

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
