## [Decision Letter · Decision Letter 0]

9 Sep 2024

PONE-D-24-27876Standard deviation: standardized bat monitoring techniques work better in some ecosystemsPLOS ONE

Dear Dr. Haelewaters,

Thank you for submitting your manuscript to PLOS ONE. After careful consideration, we feel that it has merit but does not fully meet PLOS ONE’s publication criteria as it currently stands. Therefore, we invite you to submit a revised version of the manuscript that addresses the points raised during the review process.

We look forward to receiving your revised manuscript.

Kind regards,

Lyi Mingyang, Ph.D.

Academic Editor

PLOS ONE

Journal requirements: 1. When submitting your revision, we need you to address these additional requirements. Please ensure that your manuscript meets PLOS ONE's style requirements, including those for file naming. The PLOS ONE style templates can be found at https://journals.plos.org/plosone/s/file?id=wjVg/PLOSOne_formatting_sample_main_body.pdf and https://journals.plos.org/plosone/s/file?id=ba62/PLOSOne_formatting_sample_title_authors_affiliations.pdf. 2. Your ethics statement should only appear in the Methods section of your manuscript. If your ethics statement is written in any section besides the Methods, please move it to the Methods section and delete it from any other section. Please ensure that your ethics statement is included in your manuscript, as the ethics statement entered into the online submission form will not be published alongside your manuscript.  3. Please review your reference list to ensure that it is complete and correct. If you have cited papers that have been retracted, please include the rationale for doing so in the manuscript text, or remove these references and replace them with relevant current references. Any changes to the reference list should be mentioned in the rebuttal letter that accompanies your revised manuscript. If you need to cite a retracted article, indicate the article’s retracted status in the References list and also include a citation and full reference for the retraction notice.

Reviewers' comments:

Reviewer's Responses to Questions

**Comments to the Author**

1. Is the manuscript technically sound, and do the data support the conclusions?

Reviewer #1: Yes

Reviewer #2: Yes

2. Has the statistical analysis been performed appropriately and rigorously? 

Reviewer #1: Yes

Reviewer #2: N/A

3. Have the authors made all data underlying the findings in their manuscript fully available?

Reviewer #1: Yes

Reviewer #2: Yes

4. Is the manuscript presented in an intelligible fashion and written in standard English?

Reviewer #1: Yes

Reviewer #2: Yes

5. Review Comments to the Author

Reviewer #1: In the paper “Standard deviation: standardized bat monitoring techniques work better in some ecosystems” the authors completed mist-netting surveys for bats in two widely separated field sites, Calakmul Biosphere Reserve (CBR), a Mexican lowland tropical forest, and Krka National Park (KNP), a Mediterranean dry scrub forest in Croatia. This manuscript is well organized, and the drawn conclusions are coherent with the obtained results. Despite I have enjoyed reading your paper; I feel that it needs to be corrected by a native English speaker because I have seen a few grammatical errors. I hope to provide very useful suggestions to improve the overall clarity of your study as well as the quality of your analysis. I think that my suggestions look feasible to you, and I believe you will be able to address them. Thus, please take care to do a full revision of your manuscript according to all my comments. Improvements based on my comments will be crucial for acceptance. I have some concerns and suggestions for each aspect of the manuscript. Please see below:

Lines 51 - 53: I think that you should add these important and recent references to support your sentence: “Assumptions regarding uniform performance of standardized survey methods may be problematic, given that variability in ecological and geographical factors offer challenges and present obstacles to standardized survey efficacy”. I would like to suggest:

Fraissinet, M., et al., (2023). Responses of avian assemblages to spatiotemporal landscape dynamics in urban ecosystems. Landscape Ecology, 38(1), 293-305.

Xie, B., et al., (2023). Evaluation, comparison, and unique features of ecological security in southwest China: A case study of Yunnan Province. Ecological Indicators, 153, 110453.

Lines 65 - 66: I think that you should add these important and recent references to support your sentence: “determine demographic and breeding parameters of a sampled population, to obtain samples (DNA, chemical samples, fecal samples, ectoparasites)”. I would like to suggest:

Del Vaglio, M. A., et al., (2011). Feeding habits of the Egyptian fruit bat Rousettus aegyptiacus on Cyprus island: a first assessment. Hystrix, 22(2) 2011: 281-289

Talbot, B., et al., (2017). Comparative analysis of landscape effects on spatial genetic structure of the big brown bat and one of its cimicid ectoparasites. Ecology and Evolution, 7(20), 8210-8219.

Lines 70 – 81: Please, explain in detail your hypothesis and predictions. You need to expand this section if you would want to express exactly what you want to do.

Lines 153 – 162: What is the software/tool/package used in your analysis?

Lines 202 – 211: Please expand this part of the manuscript.

Reviewer #2: The paper studied the efficacy of standardized and targeted mist-netting surveys of bats in two structurally dissimilar ecosystems. The writing of the paper is concise and well-structured, making the content easy to follow. The rationale for selecting mist-netting over other non-invasive or passive bat monitoring techniques provided in the introduction was greatly appreciated. Although mist-netting is a well-established method, providing additional details on standardized mist-netting techniques for bats or highlighting how standardized mist-netting differs from non-standardized one could enhance the reader's understanding. The readers could benefit from specific information on typical survey designs for standardized mist-netting, if such exist. Additionally, readers could benefit from in text references to existing standards, guidelines, or recommendations for mist-netting of bats, along with a comparison to your selected study design. This could provide valuable context and help in assessing the methodology used. The authors demonstrate an awareness of the study's limitations and have explicitly addressed them in the discussion. Overall, this is an engaging paper, supported by collected data from an extensive fieldwork.

6. PLOS authors have the option to publish the peer review history of their article (what does this mean?). If published, this will include your full peer review and any attached files.

Reviewer #1: No

Reviewer #2: No

---

## [Author Response · Author response to Decision Letter 0]

13 Sep 2024

Dear Dr Mingyang,

Thank you for the timely review. We were very happy to see such positive feedback from both reviewers and made changes to our manuscript in accordance with their suggestions. Below, we outline how we responded to each comment. 

Reviewer #1: In the paper “Standard deviation: standardized bat monitoring techniques work better in some ecosystems” the authors completed mist-netting surveys for bats in two widely separated field sites, Calakmul Biosphere Reserve (CBR), a Mexican lowland tropical forest, and Krka National Park (KNP), a Mediterranean dry scrub forest in Croatia. This manuscript is well organized, and the drawn conclusions are coherent with the obtained results. Despite I have enjoyed reading your paper; I feel that it needs to be corrected by a native English speaker because I have seen a few grammatical errors. I hope to provide very useful suggestions to improve the overall clarity of your study as well as the quality of your analysis. I think that my suggestions look feasible to you, and I believe you will be able to address them. Thus, please take care to do a full revision of your manuscript according to all my comments. Improvements based on my comments will be crucial for acceptance. I have some concerns and suggestions for each aspect of the manuscript. Please see below:

RESPONSE: We thank the reviewer for their positive feedback. The manuscript was written by two native English speakers. We also had it revised prior to submission by a third native English speaker, who is a professional English editor. While we are broadly happy with the quality of writing in the manuscript (see also Reviewer 2’s comments on the high writing quality), we have followed the reviewer’s advice and have carefully proof-checked the manuscript again. This has led to us making a few further grammatical corrections where appropriate (see, for example, minor edits in lines 142, 146-147, 199-200, and 244-245). 

Lines 51 - 53: I think that you should add these important and recent references to support your sentence: “Assumptions regarding uniform performance of standardized survey methods may be problematic, given that variability in ecological and geographical factors offer challenges and present obstacles to standardized survey efficacy”. I would like to suggest:

Fraissinet, M., et al., (2023). Responses of avian assemblages to spatiotemporal landscape dynamics in urban ecosystems. Landscape Ecology, 38(1), 293-305.

Xie, B., et al., (2023). Evaluation, comparison, and unique features of ecological security in southwest China: A case study of Yunnan Province. Ecological Indicators, 153, 110453.

RESPONSE: Thank you. We added these references as suggested.

Lines 65 - 66: I think that you should add these important and recent references to support your sentence: “determine demographic and breeding parameters of a sampled population, to obtain samples (DNA, chemical samples, fecal samples, ectoparasites)”. I would like to suggest:

Del Vaglio, M. A., et al., (2011). Feeding habits of the Egyptian fruit bat Rousettus aegyptiacus on Cyprus island: a first assessment. Hystrix, 22(2) 2011: 281-289

Talbot, B., et al., (2017). Comparative analysis of landscape effects on spatial genetic structure of the big brown bat and one of its cimicid ectoparasites. Ecology and Evolution, 7(20), 8210-8219.

RESPONSE: We were unable to find sufficient relevance for the first reference to be cited in our manuscript. However, we did cite the second suggested reference, as suggested by Reviewer #1.

Lines 70 – 81: Please, explain in detail your hypothesis and predictions. You need to expand this section if you would want to express exactly what you want to do.

RESPONSE: We added our two hypotheses, in lines 84-88: “We formulated two hypotheses: (1) standardized mist-netting is more effective for capturing bats in a landscape with greater heterogeneity and abundant features suitable for commuting (e.g., rivers, forest edges) than in a homogenous landscape; and (2) in homogenous landscapes, targeted surveys result in greater efficacy in capturing bats compared to standardized surveys.” 

We also refer back to these hypotheses in the discussion in lines 268-271: “In summary, for the reasons discussed above, our results closely matched the expectations laid out in our hypotheses; 1) that standardized mist-netting performed better in the (more heterogenous) habitats of CBR than in the (more homogenous) landscapes of KNP, and 2) that targeted surveys yielded better results than standardized surveys in the homogenous landscapes of KNP.”

Lines 153 – 162: What is the software/tool/package used in your analysis?

RESPONSE: We added this information, in lines 176-177: “All analyses were done in the R language and environment for statistical computing [42]. Figures were produced using ggplot() implemented in the R package ggplot2 [43].”

Lines 202 – 211: Please expand this part of the manuscript.

RESPONSE: We expanded this section by including more references and expanding the relevance of our study to similar patterns found in other taxa (i.e., birds).

Reviewer #2: The paper studied the efficacy of standardized and targeted mist-netting surveys of bats in two structurally dissimilar ecosystems. The writing of the paper is concise and well-structured, making the content easy to follow. The rationale for selecting mist-netting over other non-invasive or passive bat monitoring techniques provided in the introduction was greatly appreciated. Although mist-netting is a well-established method, providing additional details on standardized mist-netting techniques for bats or highlighting how standardized mist-netting differs from non-standardized one could enhance the reader's understanding. The readers could benefit from specific information on typical survey designs for standardized mist-netting, if such exist. Additionally, readers could benefit from in text references to existing standards, guidelines, or recommendations for mist-netting of bats, along with a comparison to your selected study design. This could provide valuable context and help in assessing the methodology used. The authors demonstrate an awareness of the study's limitations and have explicitly addressed them in the discussion. Overall, this is an engaging paper, supported by collected data from an extensive fieldwork.

RESPONSE: We appreciate these very positive comments from Reviewer #2. We have incorporated important references with regard to standardized mist-netting as suggested. We note that these references were already cited but we added some additional sentences to draw the attention to best practices and guidelines, in lines 74-77: “Best practices and guidelines for standardized mist-netting for bats are presented by Kunz and Kurta [23], Barlow [24], Walsh and Catto [19], Battersby [21], and Collins [20]. Survey effort for mist-netting has been explored in further detail by Weller and Lee [25] and Hughes and colleagues [26].”

---

## [Editor Report · Decision Letter 1]

23 Sep 2024

Standard deviation: standardized bat monitoring techniques work better in some ecosystems

PONE-D-24-27876R1

Dear Dr. Haelewaters,

We’re pleased to inform you that your manuscript has been judged scientifically suitable for publication and will be formally accepted for publication once it meets all outstanding technical requirements.

Kind regards,

Lyi Mingyang, Ph.D.

Academic Editor

PLOS ONE

Additional Editor Comments (optional):

I have read myself the new version of the manuscript and found it suitable for the publication in PlosOne.

Best regards,

LM
---

## [Editor Report · Acceptance letter]

26 Sep 2024

PONE-D-24-27876R1 

PLOS ONE

Dear Dr. Haelewaters, 

I'm pleased to inform you that your manuscript has been deemed suitable for publication in PLOS ONE. Congratulations! Your manuscript is now being handed over to our production team.

Kind regards, 

on behalf of

Professor Lyi Mingyang 

Academic Editor

PLOS ONE